# Transcriptomic Analysis on the Effects of Altered Water Temperature Regime on the Fish Ovarian Development of *Coreius guichenoti* under the Impact of River Damming

**DOI:** 10.3390/biology11121829

**Published:** 2022-12-15

**Authors:** Ting Li, Qiuwen Chen, Qi Zhang, Tao Feng, Jianyun Zhang, Yuqing Lin, Peisi Yang, Shufeng He, Hui Zhang

**Affiliations:** 1State Key Laboratory of Hydrology-Water Resources and Hydraulic Engineering, Nanjing Hydraulic Research Institute, Nanjing 210029, China; 2College of Water Resource and Hydropower, Sichuan University, Chengdu 610065, China; 3Center for Eco-Environmental Research, Nanjing Hydraulic Research Institute, Nanjing 210029, China; 4Changjiang River Scientific Research Institute, Wuhan 430010, China; 5Yangtze Institute for Conservation and Green Development, Nanjing 210029, China

**Keywords:** development of fish ovary, water temperature, transcriptome, cell division, vitellogenin accumulation

## Abstract

**Simple Summary:**

The reproductive processes of fish are known to be sensitive to water temperatures. However, the mechanism by which altered water temperature by river damming affects fish reproduction remains unclear. This study investigated the effects of altered water temperature on the development of fish gonads by differential expression of temperature-sensitive transcripts. Particularly, the regulatory mechanism of oocyte meiotic division and the accumulation of nutrient materials were explored. We found that elevated temperatures, from a low value to a physiological optimum temperature, was conducive to cell division and vitellogenin accumulation; while elevated temperatures, from physiological optimum temperature to a high temperature within a certain range, was favorable for cell division but not vitellogenin accumulation. Although a warm winter temperature favors early gonadal development, it can impair final spawning quantity and quality in spring or summer. These results provide robust evidence that under the impacts of reservoir operation, the altered water temperature regime affects the gonadal development in temperate fish.

**Abstract:**

Field investigation indicated that the reduction in fish spawning was associated with the alteration in water temperatures, even a 2–3 °C monthly difference due to reservoir operations. However, the physiological mechanism that influences the development of fish ovary (DFO) remains unclear. Thus, experiments of *Coreius guichenoti* were conducted at three different temperatures, optimal temperature (~20 °C, N) for fish spawning, lower (~17 °C, L), and higher (~23 °C, H), to reveal the effects of altered water temperature on the DFO. Comparisons were made between the L and N (LvsN) conditions and H and N (HvsN) conditions. Transcriptomic analysis differentially expressed transcripts (DETs) related to heat stress were observed only in LvsN conditions, indicating that the DFO showed a stronger response to changes in LvsN than in HvsN conditions. Upregulation of DETs of vitellogenin receptors in N temperature showed that normal temperature was conducive to vitellogenin entry into the oocytes. Other temperature-sensitive DETs, including microtubule, kinesin, dynein, and actin, were closely associated with cell division and material transport. LvsN significantly impacted cell division and nutrient accumulation in the yolk, whereas HvsN only influenced cell division. Our results highlight the impact of altered water temperature on the DFO, thereby providing insights for future reservoir operations regarding river damming and climate change and establishing fish conservation measures.

## 1. Introduction

More than 45,000 large reservoirs have been constructed globally to improve flood management and energy security [1]. However, these reservoirs significantly alter water temperature regime in rivers, particularly in tropical or subtropical regions, which is about 2–3 °C higher in winter, and lower in summer, compared to natural conditions [2,3,4]. Owing to their ectothermic nature, fish are vulnerable to temperature variations [5,6,7], particularly in the reproductive stage [8]. Therefore, the impact of water temperature changes in dammed rivers on fish reproduction has attracted much attention. For instance, the operation of reservoirs in the Yangtze River in China altered the river water temperature regime, consequently reducing the effective reproductive population of Chinese sturgeons to 0–4.5% with the alteration of the water temperature rhythm [9]. Furthermore, changes in the water temperature due to the operation of the Three Gorges Reservoir led to delayed spawning in four Chinese carps [2]. Li et al. [10] also demonstrated that fish egg quality and quantity are affected by the effective thermal accumulation during the development of fish ovary (DFO). Thus, these studies highlight the effects of altered water temperature on the DFO; however, the physiological mechanisms underlying the effects of water temperature on the DFO remain largely unclear.

DFO refers to the growth of oocytes from the primary to the secondary phase, and primarily involves the processes of oocyte meiotic division and nutrient accumulation [11,12]. Moreover, the DFO determines the number and quality of gametes and plays an important role in the fish spawning quality and survival rate [13,14]. Temperature is a crucial factor that affects oocyte meiosis during the DFO. Differentially expressed transcripts (DETs) of heat-related genes during the DFO is the key factor for us to understand the influence mechanism of temperature. Differentially expressed transcripts (DETs) of heat shock proteins (HSPs) are commonly observed in fish liver, heart, gills, kidney, and gonads in response to temperature changes [15,16,17,18]. HSPs are also reported to influence the meiotic processes, such as chromosome separation in other vertebrate oocytes [19]. The exhaustion or lack of the Hsp90 inhibits germinal vesicle breakdown in mammals (during G2/M transition), leading to delayed ovarian development [20]. Additionally, heat stress conditions cause abnormal actin filament formation and disorderly microtubule distribution in the oocyte cytoskeleton [13,21,22]. Moreover, temperature-sensitive DETs of tubules and motor proteins play an important role in separating the chromosomes during oocyte meiosis [22,23]. During meiosis, kinetochore-microtubule (tubule with temperature-sensitive DETs) and kinesin (motor protein) dynamics are associated with maintaining spindle coordination to promote chromosome separation [24]. Thus, temperature-sensitive DETs, such as those of HSPs, microtubules, and motor proteins for dynamics provide insights into the effects of temperature variation on oocyte meiosis.

Vitellogenin accumulation in oocytes is another process affected by temperature variation during the DFO. The yolk, which provides nourishment to embryonic development and larval growth, is primarily derived from vitellogenin during the DFO [25]. However, temperature affects vitellogenin formation in the liver by regulating the hormonal response through the hypothalamic-pituitary-gonad axis [26]. Vitellogenin is recognized by vitellogenin receptors (VtgRs) on the oocyte membrane [27], and it enters the oocyte via the VtgRs by endocytosis and dynamic transport [28]. Myosin, a molecular motor protein, mediates the dynamic transport of endocytic vesicles and other cargos along the actin filaments [29,30]. Similar to ATP or GTP molecules, elevated temperatures (within a certain range) also promote the activity of motor proteins [31,32]. Thus, the DETs of VtgRs and dynamic transport proteins represent another potential association between water temperature variations and the DFO.

Therefore, we explored the potential effects of altered water temperature on the DFO via two main physiological processes: oocyte differentiation (meiosis) and oocyte nutrition accumulation (yolk). The primary objective of this study is to explore the physiological mechanism underlying the effects of water temperature variations on the DFO. The results of this study highlight the importance and need for water temperature management measures for the conservation of fish in dammed rivers.

## 2. Materials and Methods

The upper Yangtze River (also known as the Jinsha River; China) is heavily regulated by cascade dams (Figure A1, Table A1), and compared to natural conditions, the operation of these reservoirs can alter the daily average water temperature by ±2–3 °C in winter and summer (Figure A2). *Coreius guichenoti* is one of the six indicative fish species in the Jinsha River which spawns once per year, typically from April to July. The field investigations revealed that the water temperature variations due to the reservoir operations severely affected the spawning abundance of *C. guichenoti* (Figure A1 and Figure A2, Table A1). The optimal flow rate for *C. guichenoti* spawning is approximately 0.5 m s^−1^ [33]. The DFO of *C. guichenoti* occurs from December to May, when the water temperature increases from a low temperature of 15 °C to the initial spawning temperature of 20.4 °C (Figure A2A). The highest temperature recorded in the spawning section of the river during field investigations was approximately 24 °C (Figure A2).

### 2.1. Experimental Setup and Process Control

To explore the effects of water temperature alteration on the DFO in *C. guichenoti*, an optimal water temperature of 20 °C, suitable for *C. guichenoti* spawning, was used as the control, and 20 ± 3 °C was used as the test range (15–24 °C). The experiment was conducted in three annular flumes (Appendix A), and the mean flow velocity was maintained at 0.5 ms^−1^. Temperatures in the three annular flumes were set at 20 ± 0.5 °C (control, N), 17 ± 0.5 °C (low, L), and 23 ± 0.5 °C (high, H), respectively, to determine their effects on the DFO in *C. guichenoti*. Experimental fish were caught by local fishermen from Xiluodu to Xiangjiaba of the Jinsha River using small hooks in November 2018. Several fish were mildly wounded and their wounds were disinfected with salt water for 5 min before captive breeding. To adapt the wild-caught fish to captivity and forage fish feeding, the fish were adaptively cultivated at a depth of 10 m in the Xiangjiaba reservoir area of the Jinsha River for nearly six months prior to the experiments. Considering that the sexes of *C. guichenoti* are indistinguishable, the fish were randomly divided into three experimental groups (*n* = 15/group; healthy; weight: 116–121 g) with natural male ratios. Three samples per group were used as biological replicates for mRNA extraction and sequencing [34].

During the experiment, a low-noise flow rate controller (Hangkai brand, Taizhou city, China) for each annular flume made of cement (total volume ~2.8 m^3^; water volume: ~1.8 m^3^) was used to drive the water flow to avoid disturbing the fish. Electric heating rods (800 W; *n* = 5/group; ±1.0 °C accuracy, Yee brand, Weifang city, China) were used for heat control, and water chill (5.28 kW) was used for cooling control and emergency temperature control. Temperature alarms (*n* = 2/group; ±0.5 °C accuracy, Suhed brand, Shenzhen city, China) were used for constant monitoring of the water temperature. To maintain the water temperature and avoid exposure to direct natural light, the sink was covered with semi-transparent tarpaulin. Small oxygen pumps (*n* = 1/group, 110 W, Yonglin brand, Hangzhou, China) were used continuously to ensure sufficient oxygen content in the water. The fish was fed forage in the flume every day at 18:00, and the propeller was turned off for 2 h. Experimental annular flumes were built along the Jinsha River, and the water for the experiment was obtained from the Jinsha River. One-third of the water body was replaced, and the residue was exchanged every 5 days. After the adaptive cultivation of the fish for six months, most fish maintained a good physical condition throughout the experimental period, with less than three casualties recorded per experimental group.

Since monthly temperature alterations were observed under reservoir operation (shown in Figure A2A), the experiment lasted for 30 days. When the experiment started in May 2019, the water temperature of the cage culture with a depth of 10 m was about 17 °C. Thereafter, the fish were anesthetized using 100 mgL^−1^ tricalcium methanesulfonate (MS-222) and euthanized. The fish were then weighed, and their ovaries were obtained by dissection. One of the ovaries of each fish was immobilized in Bouin’s solution (Shanghai Kehui Biotechnology, Shanghai, China) for histological examination. The other ovary was washed with diethylpyrocarbonate solution, sealed in a 1.5 mL sterile centrifuge tube (DNA–RNA free), and individually fixed in Bouin’s solution (Shanghai Kehui Biotechnology). The samples were immediately stored in liquid nitrogen until further analysis.

### 2.2. Histological Examination

The ovary samples were fixed in Bouin’s solution (Shanghai Kehui Biotechnology) before slicing them into sections on a microtome (Leica, Wetzlar, Germany). Thereafter, the samples were dehydrated, embedded in paraffin, and stained with eosin and hematoxylin (H&E) for microscopic observation (Nikon eclipse ni-u, Tokyo, Japan). The images were recorded using CaseViewer_2.0__RTM software (version 2.0.2.61392).

### 2.3. Transcriptomic Analysis

The PacBio full-length transcriptome was sequenced in a circular manner by using single-molecule and real-time (SMRT) sequencing technology to construct SMRT cell libraries. Owing to its long read length, PacBio sequencing was selected for structural correlation analysis of transcripts; however, it is expensive and cannot quantify mRNA expression. Therefore, Illumina sequencing was used to compare the Differentially expressed transcripts (DETs) in the samples. For non-model species, we used low-coverage PacBio SMRT data to improve the integrity of Illumina’s genome to reduce error by direct assembly. The detailed relationship between PacBio and Illumina transcriptome analysis is presented in Appendix A.

#### 2.3.1. Library Construction and PacBio Sequencing

The SMARTer™ PCR cDNA Synthesis Kit (Takara, Shiga, Japan) was used to synthesize full-length cDNA from sample mRNA. After PCR amplification, quality control, and purification, the library was run on the BluePippin Size Selection System (Sage Science, Beverly, MA, USA) for size selection. Thereafter, a library of 0–1, 1–2, 2–3, 3–6, and 5–10 kB was constructed. Lastly, SMRT cells were sequenced using P6-C4 chemical reagents on a PacBio RS II platform.

#### 2.3.2. Library Construction, Illumina Sequencing, and Transcriptome Assembly

Nine samples (*n* = 3/group) were used for Illumina sequencing. Illumina libraries were prepared using the NEBNext Ultra™ RNA Library Prep Kit (E7530L; New England Biolabs, Ipswich, MA, USA) as follows: (1) The mRNA was enriched using oligo (dT) magnetic beads; (2) mRNA was randomly fragmented using fragmentation buffer; (3) the first cDNA strand was transcribed from the mRNA template using random hexamers, and the second cDNA strand was synthesized using a mixture of buffer, dNTPs, RNase H, and DNA polymerase I; (4) AMPure XP Beads (Beckman, Brea, CA, USA) were used to purify cDNA strands; and (5) the double-stranded cDNA was repaired, poly A-tailed, and sequenced. AMPure XP Beads (Beckman) were used for fragment size selection. The amplified mRNA library was sequenced on an Illumina HiSeq X Ten platform (San Diego, CA, USA) to produce double-ended sequences (~150 bp). The original short fragment was quality-filtered using NGSQC Toolkit v2.3.324 [35] after removing the first five bases from the 5’ end.

#### 2.3.3. PacBio Transcriptome Quality Control and Analysis

For the PacBio quality control, the raw polymerase-read sequences with a length of <50 bp or a sequence accuracy <0.90 were filtered, and the adapters were removed to obtain the subreads. Lastly, subreads with a length of <50 bp were filtered to obtain clean data for further analysis.

The process for obtaining the full-length transcriptome primarily consists of the following steps: (1) full-length sequence identification, (2) isoform level clustering to obtain a consistent sequence, and (3) uniform sequence polishing. All the raw sequences were converted to circular consensus (CCS) reads according to the connector in the sequence (adaptor), and the sequence was divided into full-length sequences and non-full-length sequences based on whether they contained 3′ primers, 5′ primers, or Poly A tail. The full-length sequences from the same transcript were clustered, and a consistent sequence was obtained from each cluster. High-quality sequences were polished for subsequent analysis. While the low-quality CCS were corrected by Illumina short reads with the LoRDEC tool v0.6 with -k 21 and -s 3, default settings were used for the other parameters [36], as shown in Appendix A. A reference database named OrthoDB v2.8.0 (https://www.orthodb.org/ accessed on 30 December 2020) was used to evaluate the integrity and accuracy of the transcripts via BUSCO [37].

#### 2.3.4. Illumina Transcriptome Quality Control and Analysis

The quality of Illumina data was controlled using Cutadapt software (https://github.com/marcelm/cutadapt/ accessed on 30 December 2020) at default parameters [38]. We removed the adapter and primer sequences from the reads and then filtered the low-quality data to obtain clean high-quality reads for subsequent analysis.

By using STAR v 2.5.3a with default options, clean reads for Illumina were mapped to PacBio reads to reduce the large error caused by de novo assembly, and quantified expression levels as fragments per kilobase of transcripts per million mapped reads (FPKM) using RSEM tool v1.1.11 (without assembly) at default parameters [39], as shown in Appendix A. For non-model species, low-coverage PacBio SMRT data were used to improve the integrity of Illumina’s genome and to reduce errors caused by direct use of the second-generation assembly. DESeq2 was used to analyze the RNA-seq data [40]. DETs were identified as those with a false discovery rate (FDR) of ≤0.05 and a fold-change ≥ 2. FDR was obtained using the Benjamini–Hochberg method to a corrected significant *p*-value. DETs generated in low temperature versus high temperature (LvsN) groups and high temperature versus low temperature (HvsN) groups were annotated using the zebrafish genome database (version GRCz11.103) to determine the name and function of the differentially expressed genes for subsequent analysis. The Metascape software (https://metascape.org/gp/index.html#/main/step1 accessed on 30 October 2020) was used for Gene Ontology (GO) term clustering analysis of DETs at default parameters.

## 3. Results

### 3.1. Morphological Characteristics of Ovarian Tissues

Figure 1 shows the development of ovaries in *C. guichenoti*. The appearance of the ovarian tissues of each group is relatively similar under naked eye observation, as shown in Figure 1A; however, oocytes were varied in size, irregularly shaped (nearly round), and in close proximity to each other (Figure 1B). The diameters of the largest oocyte and the nucleus were approximately 128 and 57 μm, respectively (Figure 1C). The variation in the oocyte sizes indicates that the oocytes underwent asynchronous development. Yolk nuclei were clearly visible in the oocyte nucleus. Furthermore, macroscopic and microscopic observations of the ovaries revealed that, based on morphology, the ovarian tissue was in stage II [11]. There was no apparent difference in the gonads among the different temperature groups during the one-month control period; therefore, we conducted the transcriptome assay to reveal the intrinsic effect of water temperature at the molecular and cellular levels.

### 3.2. Transcriptome Data and Their Quality

PacBio sequencing of the full-length reads and quantitative Illumina sequencing were performed to reduce errors of the assembly sequencing from non-model species. The cleaned PacBio sequencing data of 24.41 and 60.4 Gb from the nine samples were obtained.

A total of 291,134 CCS reads, including 246,459 full-length non-chimeric sequences, were obtained via PacBio sequencing. The length distribution of CCS sequence libraries obtained from the cDNA is shown in Figure 2A. The integrity analysis of the transcripts shows >79% coverage (Figure 2B). After removing the redundant CCS sequences, 36,734 sequences were annotated, of which 44% were fully covered, 32% were single copies, 3% were multiple copies, and 21% were mismatched.

Statistical evaluation of Illumina sequencing and mapping efficiencies are shown in Table 1. The GC content of the sequenced bases was 48.39–48.90%, and its Q30 (99.9% raw sequencing accuracy) was greater than 94.92%, indicating high sequence quality. The non-redundant transcript mapping results of Illumina and PacBio sequencing showed that the uniquely mapped, multiple mapped, and unmapped reads account for 38–42%, 43–47%, and 16–17%, respectively, of the total reads.

### 3.3. DETs and Their Functional Enrichment

A comparison of the DETs identified under different temperature conditions is shown in Figure 3. A total of 545 DETs were identified in the LvsN conditions, among which 273 were upregulated and 272 were downregulated. In contrast, a total of 393 DETs were identified in HvsN conditions, among which 224 were upregulated and 169 were downregulated. The number of DETs in the LvsN condition was greater than that in the HvsN condition, indicating that the effect of water temperature was stronger during the DFO under LvsN conditions. As *C. guichenoti* is a non-model species, its DETs were annotated using the zebrafish gene bank to determine the gene names (version GRCz11.103), and 407 and 295 DETs were annotated for the LvsN and HvsN conditions, respectively. Certain annotated transcripts were commonly upregulated or downregulated between the test groups; 36 transcripts were common between the upregulated transcripts in the LvsN condition (LvsN_up) and the downregulated transcripts in the HvsN condition (HvsN_down).

The GO enrichment terms of the annotated genes corresponding to the DETs are shown in Figure 4. In the LvsN comparison (Figure 4A), functions such as cell cycle process, transferase complex, nucleoplasm, mRNA processing, cell cycle phase transition, protein import into nucleus, chromosomal region, and microtubule cytoskeleton were enriched. In contrast, in the HvsN comparison (Figure 4B), intracellular protein-containing complex, oogenesis, nucleolus, supramolecular complex, and nuclear localization of proteins were enriched. Therefore, in both LvsN and HvsN, the common enriched functions involved various regulatory processes associated with oocyte development, including cell cycle, oogenesis, chromosome and microtubule organization, and mRNA processing.

### 3.4. Key DETs Involved in Oocyte Development Are Affected by Temperature

Considering the influence of temperature and gene function enrichment on ovarian development, temperature-sensitive DETs of HSPs, yolk receptor, tubulin, and motor-related functions were analyzed, and the expression levels and functional annotations of the DETs are shown in Figure 5 and Appendix A, respectively.

Temperature variations affected the expression of HSPs (Figure 5A). The DETs of HSPs were only expressed in LvsN and not in HvsN conditions. *hsp90ab1* and its regulatory factor *hsf1* (Appendix A), associated with heat response and cell proliferation (Appendix A), were downregulated in LvsN conditions. Additionally, the response of *C. guichenoti* ovarian tissue to heat stress was significantly higher in the LvsN than the HvsN conditions.

Temperature variations also affected the DETs of VtgRs and endocytic transport (Figure 5B). The VtgR-related transcripts, *ldlrap1a*, *lrp13*, and *loc100536757* were downregulated under both LvsN and HvsN conditions. Additionally, *picalm* and *arpc5lb* (subunit of the *arp2/3* complex), which play an important role in mediating clathrin endocytosis (Appendix A), were downregulated under the LvsN condition. Collectively, receptor-associated transcripts required for vitellogenin entry into cells showed higher expression at N, compared to L and H temperatures.

The DETs of the microtubule are shown in Figure 5C. *tuba4l*, *tubb2b*, and *mapta* were downregulated under HvsN conditions, while *tuba1b* was upregulated in LvsN conditions. *tuba4l*, *tubb2b*, and *tuba1b* are the main constituents of tubulin (Appendix A). Similarly, microtubule-related transcripts showed lower expression levels at high temperatures.

DETs of motor-related proteins are shown in Figure 5D. The differential transcript expression level of *kifs*, at a fragments per kilobase million (FPKM) range, was 0–11. *kif15* and *kif5c* were upregulated, whereas *kif4* was downregulated in LvsN. In contrast, *kif20b* and *kifap3b* expression was upregulated in HvsN. Furthermore, *dync1i2a* was downregulated in LvsN, whereas *dctn1a* was upregulated in HvsN, indicating that elevated temperature increases the expression level of dynein. Actin-related transcripts, *alan* and *actb1*, were upregulated at N temperature. The DETs level of *actb1* was high, with an FPKM range of 39–143. The myosin transcripts, except for *mybph* and *mylk5* were upregulated under LvsN and HvsN conditions, whereas other transcripts, such as *myh9b*, *myh9a*, *myho1b*, and *myo10l3*, were downregulated in LvsN and HvsN conditions. These results suggest that the expression levels of the DETs of actin and myosin were higher under N than under L and H temperature conditions.

Based on the DETs in the LvsN and HvsN conditions, we proposed a mechanism in Figure 6 to describe the impact of water temperature regime alterations (incurred by river damming) on the DFO, which consequently influences fish reproduction.

## 4. Discussion

### 4.1. Water Temperature Is a Key Factor Affecting Fish Spawning

The reproductive stage is the most sensitive and vulnerable stage in fish that is considerably affected by external environmental factors. Variations in the flow velocity [41], water temperature [42], and turbidity [43] affect the three main stages of reproduction, namely induction (the beginning of oogenesis), vitellogenesis (yolk formation), and maturation (including ovulation and oviposition) in the temperate fish [44]. The spawning time of the fish is closely associated with the water temperature regime. Some representative fish, such as Alaska pollock, Atlantic cod, Common sole, Pacific cod, Burbot, European perch, and American yellow perch require water temperatures < 18 °C to spawn, and elevated temperatures promote oviposition; whereas, other representative fish, such as Atlantic salmon, *Gobiocypris rarus*, and Argentinian silverside, require water temperatures > 18 °C to spawn, and elevated temperatures delay or inhibit oviposition [7]. In addition, temperature is a decisive factor in the reproductive development (vitellogenesis) of temperate fish, and vitellogenesis occurs faster at high temperatures [44].

In this study, variations in factors, such as water velocity, turbidity, and temperature, due to reservoir operation, potentially influence *C. guichenoti* reproduction. However, different aged fish and spawning grounds of *C. guichenoti* are distributed 0–60 km below the Xiluodu dam, as shown in Figure A1. Moreover, the turbidity may be low in clear water, under reservoir interception, due to a decrease in sediment content [45]. Therefore, unlike temperature variations, changes in velocity and turbidity may not be the main reasons for the absence of spawning behavior during the historical spawning period.

### 4.2. Effects of Water Temperature on the Differentiation in DFO

Organisms often have an optimal temperature zone, at which the environmental temperature is conducive to physiological functions [46,47]. Compared to homothermic animals, with a temperature fluctuation range of ±0.5 °C [48], heterothermic animals show increased tolerance to temperature variations.

The HSPs family is widely used to characterize the response of organisms to temperature variations [49,50]. The expression of HSPs is closely associated with both heat stress and oocyte development. In mammalian meiosis, the reduction or depletion of *hsp90* blocks germinal vesicle breakdown and delays ovarian development [51]. Studies on black tiger shrimp, *Penaeus monodon*, showed that the expression level of *hsp90* varies at different stages of ovarian development, and that it is upregulated following heat treatment, particularly during the DFO [52]. In this study, the HSPs transcripts of *hsf1* and *hsp90ab1* were downregulated in LvsN; however, they remained unchanged in HvsN conditions. This may be attributed to the temperature response interval for heat stress during the DFO. *hsp90ab1* is associated with reproductive processes and cell proliferation [53,54]. Thus, upregulation of *hsf1* and *hsp90ab1* within a certain range tends to promote meiosis.

Temperature affects the expression of tubulin and other motor proteins, thereby affecting oocyte meiosis. Microtubules are assembled from tubulin subunits and are essential for chromosome separation during meiosis [24]. Oocyte meiosis requires microtubules, actin filaments, and chromosome rearrangement [55]. A study on *Caenorhabditis elegans* showed that the depolymerization and polymerization rate of the microtubules reduce with a decrease in temperature from 37 °C to 10 °C; however, the depolymerization rate is slower than the polymerization rate, leading to a decrease of tubule polymers (Li and Moore 2020 [56]). Additionally, cytoplasmic dynein and kinesin are microtubule-based molecular motors with temperature-sensitive functions [29]. In this study, microtubule-related transcripts (*tuba4l*, *tubb2b*, and *mapta*) were downregulated in the HvsN conditions, whereas *tuba1b* was upregulated in the LvsN conditions. The microtubule formation rate is not only sensitive to temperature but also positively correlated with the tubulin concentration [56]. Thus, increased expression of microtubules may compensate for polymer loss caused by the temperature reduction. Kinesin (*kif15* and *kif20b*) and dynein (*dync1i2a*) have been reported to attach to the spindle and microtubules during meiosis and play a dynamic driving role in separating the cytoplasm and chromosomes [40,57]. Kinesin and dynein were upregulated under the HvsN conditions, suggesting that high temperature is beneficial to the aggregation of microtubules and spindle formation during meiosis. Overall, the DETs of HSPs, tubulin, kinesin, and dynein at different temperatures showed that elevated temperature was conducive to chromosome-related activities during meiosis in the DFO.

### 4.3. Effects of Water Temperature on Vitellogenin Accumulation

Another mechanism underlying the temperature response in the DFO includes changes in the VtgRs and cellular transport systems that allow nutrients to accumulate in the ovaries. Although the direct effects of temperature on VtgRs protein expression have not been clearly reported, the indirect effect of temperature on vitellogenin production, via the brain-pituitary-gonad axis, has been widely recognized [12,25]. Körner et al. [58] reported that vitellogenin synthesis is often accelerated following elevated temperature, and an increase in serum vitellogenin is accompanied by elevated temperature [58,59]. Additionally, high vitellogenin concentration in the blood is accompanied by high expression of receptors in oocytes. However, vitellogenin synthesis can be limited when the temperature rises above the physiological optimum temperature [60]. Vitellogenin in the blood is recognized by VtgRs on the cell membrane and then transported into the cell by endocytosis and motor proteins [27,61]. Myosin, which travels along microfilaments, is a key motor protein involved in vesicular transport [62] and positively correlates with motor protein activity. Even temperature increase within a mild range will benefit endocytosis and vitellogenin transport [63]. Overall, elevated temperature below physiological optimum temperature is conducive to yolk accumulation.

In this study, the DETs of low-density lipoprotein receptor, *dlrap1a*, and *loc100536757* in the LvsN condition and *lrp13* in the HvsN condition were downregulated. This may be due to the temperature alteration, considering that temperatures below the physiological optimum cause accumulation of vitellogenin, whereas temperatures above the physiological optimum limit vitellogenin production [63]. Furthermore, *picalm* and *arpc5lb*, which mediate endocytosis [64,65], were highly expressed at the N temperature. Additionally, myosin, involved in transport, was downregulated in the LvsN and HvsN conditions. These results demonstrate that the N temperature was optimal for the transport of intracellular substances compared with the L and H temperatures.

### 4.4. Potential Impact of Variations in Water Temperature on Fish Reproduction in Dammed River

Flow rate and thermal regimes in rivers are the key factors influencing fish spawning and are substantially altered by reservoir operations. Previous studies have focused on the relationships between fish spawning and flow velocity as well as water temperature [3,66,67,68]; however, the effects of water temperature alterations during the DFO have not been well investigated. DFO plays a decisive role in fish reproduction [14], and most temperate fish undergo ovary development in the winter, ovary maturation in the spring, and spawning in the summer. The operation of reservoirs in tropical and subtropical regions often results in a warmer water temperature in winter [3,4], thus affecting the DFO and consequently the spawning activity [10]. This study provides physiological evidence that elevated water temperature can affect the DFO by altering meiosis and vitellogenin accumulation. For temperate fish, water temperature elevated by 2–3 °C (i.e., LvsN) is conducive to promoting oocyte differentiation and vitellogenin accumulation. This mechanism conforms to the phenomenon that an increase in low temperature leads to ovary precocity, which can disrupt the distribution mechanism of nutrients and the maturation time of fish eggs [6,69,70]. When the ambient temperature exceeds the physiologically optimal temperature (i.e., HvsN), meiosis increases in the oocytes, whereas yolk accumulation is constrained. Thus, an increase in temperature (above the optimal temperature) impairs vitellogenin production and ultimately results in reduced egg size [71,72].

In addition to the water temperature alteration caused by the reservoir, the impact of climate change on water temperature should be considered. Temperature increases of 5 °C during the winter in the North Sea can delay ovary maturation of the Lesser sand eel (*Ammodytes marinus*) (spawning in spring) by 2 months [72]. Additionally, a warmed winter caused by climate warming was reported to decrease yolk production and shorten the spawning period of *P. flavescens* [71]. Meanwhile, other studies have suggested that a short period of low temperature during the DFO has a negative effect on fish spawning quality [10,71,73]. Thus, the adverse effect of elevated water temperature on the DFO, under climate change and cascade dam development, should be considered to better protect fish resources. The cumulative effect of water temperature on the DFO is reflected not only in the DETs, but also in the hormonal regulation of other tissues [26]. Fish hormone regulation and multi-omics approaches should be used to reveal the relationship between the DFO and water temperature alterations to guide aquatic biological conservation.

## 5. Conclusions

Reservoir operations alter water temperature by 2–3 °C, thereby affecting the DFO. The DETs of HSPs, VtgRs, tubulin, and transport proteins are closely associated with temperature changes and oocyte differentiation (meiosis) or growth (vitellogenesis) during the DFO. Elevated temperature, below the physiologically optimal range, is conducive to cell division and vitellogenin accumulation. However, increased temperature, above the physiologically optimal range, is favorable for cell division but not vitellogenin accumulation. Thus, although warm winter temperature is beneficial for the DFO, it may impair final spawning quantity and quality. Therefore, considering the effects of variations in water temperature on the DFO, it is necessary to establish conservation measures to alleviate the adverse impacts of reservoir operation and climate change on fish reproduction.

## Figures and Tables

**Figure 1 biology-11-01829-f001:**
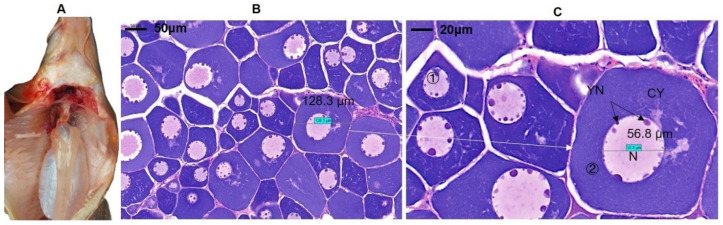
Development of ovaries in *C. guichenoti*. (**A**) Gonad appearance; (**B**,**C**) are microscopic characteristics of ovarian slices. ① Primary growth oocytes; ② large oocyte; YN: yolk nucleus; N: nucleoli; CY: cytoplasm.

**Figure 2 biology-11-01829-f002:**
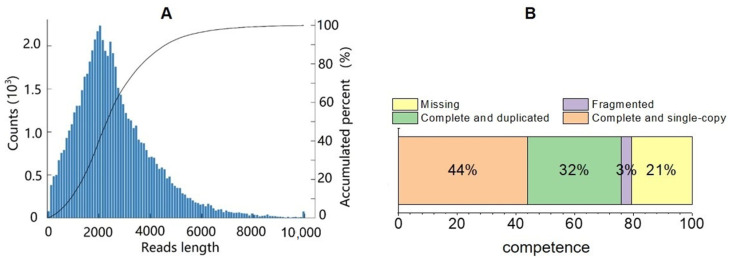
Statistical evaluation of PacBio sequencing. (**A**) length distribution of PacBio sequence. (**B**) Integrity assessment of the redundant transcripts in PacBio sequencing.

**Figure 3 biology-11-01829-f003:**
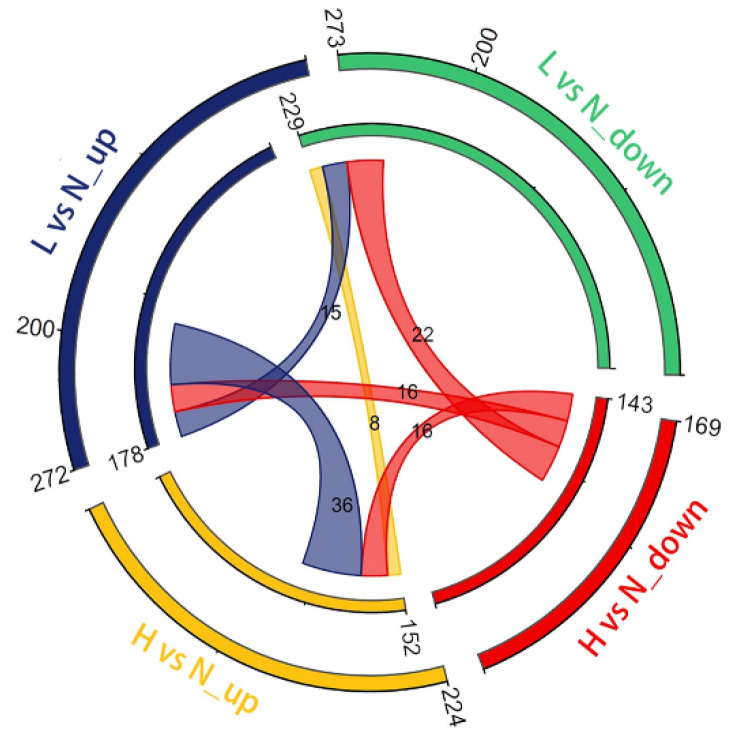
Relationship of upregulated or downregulated DETs. The outer and inner circles represent the DETs and the annotated genes of the DETs, respectively. The colored bands in the middle represent shared genes between different comparisons. The “_up” and “_down” suffixes after the comparison conditions indicate upregulated and downregulated DETs, respectively.

**Figure 4 biology-11-01829-f004:**
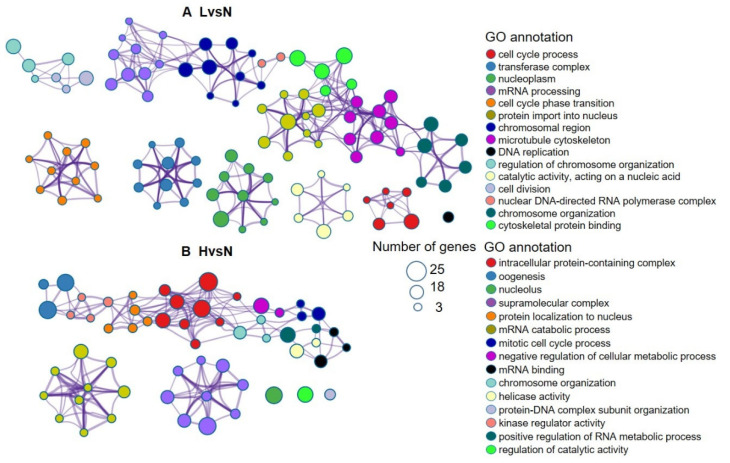
Gene Ontology (GO) enrichment terms of the annotated genes. (**A**) GO enrichment for comparisons condition between the L and N (LvsN), and (**B**) comparisons conditions and H and N (HvsN) conditions. The degree of GO enrichment term is determined on the basis of gene number and p-value. GO annotation is arranged according to the increasing order of *p*-value, and all GO annotations showed <0.01 *p*-values. A higher degree of enrichment indicates a higher rank of GO functional annotation and is represented as a larger circle.

**Figure 5 biology-11-01829-f005:**
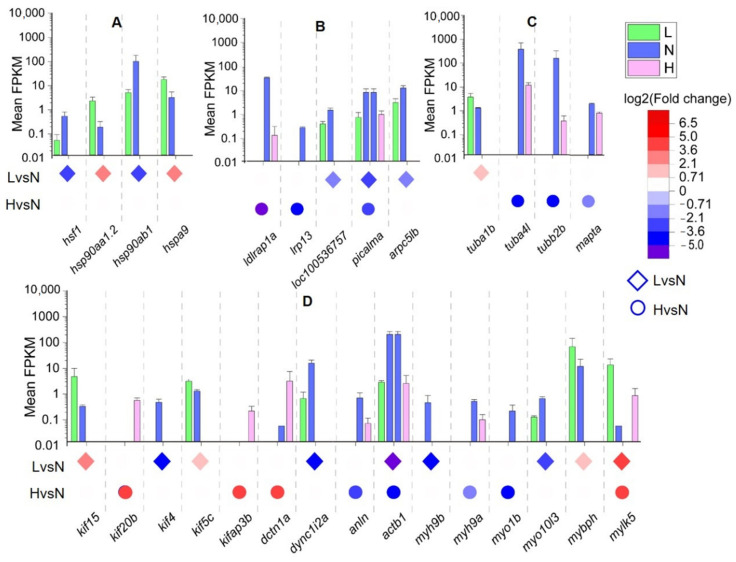
Expression level and relationship between upregulated and downregulated key DETs. The circle and square represent the fold-change in the expression of DETs in the LvsN and HvsN temperature conditions, respectively. The DETs of (**A**) heat stress, (**B**) VtgRs and endocytic transports, (**C**) microtubule, and (**D**) motor proteins. Note: The expression levels of DETs are indicated with the gene names for the LvsN and HvsN comparisons (*n* = 3).

**Figure 6 biology-11-01829-f006:**
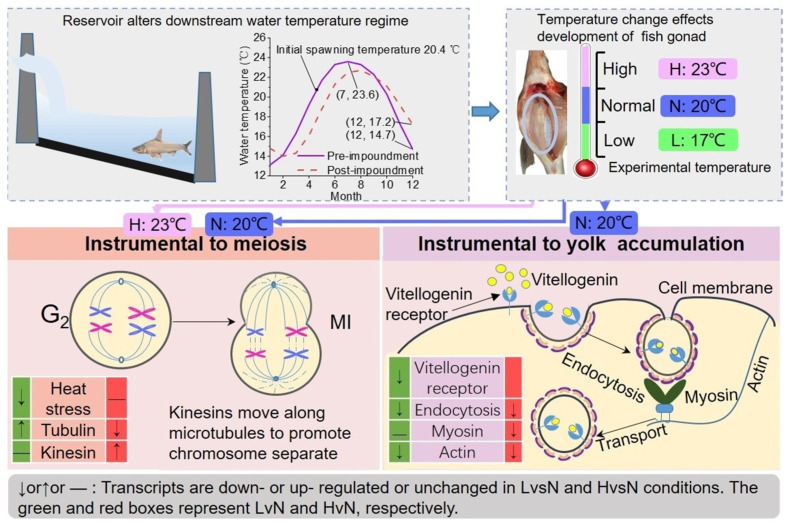
Physiological mechanism of ovary development affected by alterations in water temperature regime in the dammed river.

**Table 1 biology-11-01829-t001:** Evaluation statistics of Illumina results and mapping efficiencies (*n* = 3).

Samples	Total Clean Reads Number	GC Content (%)	≥Q30 (%)	Uniquely Mapped Reads	Multiple Mapped Reads	Unmapped Reads
N1	2.70 × 10^7^	48.70	95.04	39.11%	43.67%	16.71%
N2	2.24 × 10^7^	48.39	95.01	38.29%	42.48%	16.57%
N3	2.38 × 10^7^	48.80	94.92	38.94%	43.24%	16.80%
L1	2.39 × 10^7^	48.72	95.45	40.44%	45.41%	16.10%
L2	2.13 × 10^7^	48.89	95.07	41.22%	46.96%	16.95%
L3	2.06 × 10^7^	48.97	95.44	38.82%	43.64%	16.61%
H1	2.60 × 10^7^	48.90	94.94	39.51%	44.42%	16.10%
H2	2.42 × 10^7^	48.90	95.06	38.76%	43.48%	16.50%
H3	2.76 × 10^7^	48.87	94.96	38.09%	43.15%	16.65%

Note: Uniquely mapped reads represent the percentage of Illumina reads that were mapped to one location on the PacBio sequencing. Multiple mapped reads represent the percentage of Illumina reads that were mapped to multiple locations on the PacBio sequencing. The sequencing depth of the Illumina sequencing is higher than that of the PacBio sequencing, thus some data, with relatively low expression levels, can be sequenced by Illumina sequencing, and this part of the reads is termed as unmapped in the PacBio sequencing.

## Data Availability

The Illumina and PacBio raw data for transcriptome sequencing have been deposited in the NCBI SRA database under accession SUB7211202, PRJNA623717 (The bioprojects of HV1, HV2, HV3, NV1, NV2, NV3, LV1, LV2, LV2 in NCBI represent H1, H2, H3, N1, N2, N3, L1, L2, and L3 in Table 1).

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
