# Peer review of "Transcriptomic Analysis on the Effects of Altered Water Temperature Regime on the Fish Ovarian Development of Coreius guichenoti under the Impact of River Damming"

_biology, 2022, doi:10.3390/biology11121829_

Round 1

Reviewer 1 Report

This manuscript titled "Transcriptomic analysis on the effects of altered water tempera-2 ture regime by river damming on the development of fish 3 ovary in Coreius guichenoti" investigated the influence of altered water temperature on the  reproduction in fish. It provided the proof that the altered water temperature, as a main effect of river damming, have influenced the gonadal development in temperate fish. This result is conducive to resevior operations regarding river damming and establishing fish conservation measures.

Line 353-354, some representative fish should be specific. 

Author Response

  • We are indeed grateful to the reviewer for the positive comments and valuable suggestions that have enabled us to improve the manuscript. The specific comments are all addressed in the revised manuscript. Detailed explanations are provided below.

Reviewer 2 Report

In this manuscript, Li et al first conducted a field investigation to prove that the reduction in fish spawning was associated with the changing water temperatures altered by river damming. Then the author generated new long and short transcriptomic data from the ovaries of Coreius guichenoti cultured under three different temperature conditions. Analyzed via bioinformatic methods, the author showed that altered water temperature can affect the DFO via meiosis and yolk. The design of the work is straightforward and clear. I also agree that the relationship between fish DFO and river damming is an important issue in conservation biology. However, I have some questions and comments regarding the transcriptomic analysis. Their intention is to help the authors improve the manuscript further. They can be addressed through in-text discussions and maybe supplementary data analysis.

The title: After I read the whole paper, I think something like, “Transcriptomic analysis on the effects of changed water temperature regime altered by river damming on the development of fish ovary in Coreius guichenoti” might be better.

Ln 19: “Our results suggest” to “We found”

Ln 26: This sentence is not clear. You mean: “Field investigation indicated that the reduction in fish spawning was sensitive to water temperatures (even 2-3°C difference caused by reservoir operations).”?

Ln 27, 28: What is the meaning of (1) and (2)?

Ln 32: You should make the abbreviations clear at the first appearance of them. Like the LvsN, the N group, HvsN. Please note this throughout the paper.

Ln 46: “However, these reservoirs, significantly alter water ...”. This sentence is hard to understand. Maybe something like, “However, these reservoirs significantly alter the water temperature regime of the rivers, especially in tropical or subtropical regions, which is about 2-3°C higher in winter and lower in summer compared to natural conditions.”.

Ln 65: There should be an articulated sentence after “Temperature is a crucial factor that ...” to connect, or it will be inconsistent logic.

Ln 65, 67, 69: If you mean proteins, you should use “heat shock proteins (HSPs)” and “Hsp90”. Please use lowercase italics only for gene names. Please note this throughout the paper.

Ln 91: You don’t have to hypothesize them. You can directly list the two aspects of your study and your brief results.

Ln 98-109: I think this part is better for Results instead of Materials and Methods.

Ln 237: I think this is an interesting phenomenon. Why there were no significant histological differences? You can discuss this a little bit.

Ln 263: For Table 1, for reproducibility, you should mention which software you use to map reads.

Ln 289-297: Please only mention the important pathways you want to discuss.

Ln 325-336: This is a major concern for me. Since the majority conclusion of this paper is drawn on the expression level of the key transcripts. I think qRT-PCR analysis will make the conclusion more solid. Namely, the author can randomly select several key DEGs to confirm the reliability of the RNA-seq.

Ln 404-430: There are two many literature reviews here (Ln 404-421), which will make this part more like an Introduction and repetitive. Please try to combine these reviews with your findings organically.

Author Response

(The authors gave the same response as above.)

Round 2

Reviewer 2 Report

The author has addressed all the concerns and revised the MS accordingly. I agree with the publication of this MS. Only a minor mistake below:

Ln 27 - Please change to "monthly differences"

Author Response

We are indeed grateful to the reviewer for the positive comments and valuable suggestions that have enabled us to improve the manuscript.
